# Fundamental limits on nonequilibrium sensing

Andreas Dechant [1] & Eric Lutz[2]

The performance of equilibrium sensors is restricted by the laws of equilibrium thermodynamics. Here, we investigate the physical limits on nonequilibrium sensing in bipartite systems with both reciprocal and nonreciprocal couplings. We show that one of the subsystems, acting as a Maxwell demon, can significantly suppress the fluctuations of the other subsystem relative to its response to an external perturbation. The importance of nonreciprocal interactions for such negative violations of the fluctuation-dissipation relation to occur is identified. We further demonstrate that these violations can considerably improve the signal-to-noise ratio above its corresponding equilibrium value, allowing the subsystem to operate as an enhanced sensor. In addition, we find that the nonequilibrium signal-to-noise ratio of linear systems may be arbitrarily large at low frequencies after proper parameter optimization, even at a fixed overall amount of dissipation. These results indicate that highly accurate nonreciprocal sensors can be designed at a finite energetic cost.

Sensing plays a pivotal role in science and technology. By monitoring changes in the surroundings and reacting to external signals, sensors provide essential information about the environment of a system[1–3]. However, unwanted stochastic fluctuations fundamentally limit the amount of information that can be acquired. A sensor should provide a strong response to a signal and, at the same time, be minimally affected by the detrimental influence of noise. An important figure of merit that quantifies this property is the signal-to-noise ratio which describes how good a detected signal can be distinguished from the noise[1–3]. Recent studies of biochemical networks, in particular of the sensing of chemical concentrations by biological cells, have revealed that the breaking of detailed balance away from equilibrium can enhance sensing performance[4–11]. These findings suggest that operating sensors far from equilibrium can be of significant advantage. Yet, the fundamental physical limits on nonequilibrium sensing are still unknown[12,13].

We here address this central issue in the context of Maxwell's demon[14,15], using the tools of information thermodynamics[16,17]. By measuring a system and applying feedback, Maxwell's demon is able to extract work from an equilibrium heat reservoir by breaking the fluctuation-dissipation relation that connects the response to an

external field to the equilibrium correlation function of spontaneous fluctuations[18,19]. We show that the demon may also enhance the sensing ability of the system by exploiting their nonreciprocal interaction. Akin to the asymmetric conductance of a diode, nonreciprocity allows to strongly suppress the random fluctuations of the sensor, at the expense of those of the demon. As a consequence, the nonequilibrium signal-to-noise ratio may not only be improved compared to the equilibrium situation, it can be arbitrarily large at low frequencies in linear systems after optimization, even at a fixed overall amount of dissipation. This result implies that there is actually no fundamental limit on out-of-equilibrium sensing.

## Results

We concretely consider a generic composite system whose state space can be divided into two distinct subsystems that interact with each other. This setup acts as an autonomous Maxwell demon where one subsystem generates information and the other one reacts to it[20–25]. It additionally provides a general model for molecular sensors and two-component molecular machines that operate without external measurement and feedback[26]. When the composite system is in a nonequilibrium steady state, created for instance by nonconservative

[1]Department of Physics #1, Graduate School of Science, Kyoto University, Kyoto, Japan. [2]Institute for Theoretical Physics I, University of Stuttgart, Stuttgart, Germany. e-mail: andreas.dechant@outlook.com; eric.lutz@itp1.uni-stuttgart.de

forces, entropy is dissipated and detailed balance is broken. In the following, we combine a newly derived local form of the Harada-Sasa relation, that relates the dissipated heat to violation of the fluctuation-dissipation relation[27–30], and the second law of information thermodynamics, that extends the entropy balance to include the contribution of the information flow between the subsystems[21–25]. We show that fluctuations of one subsystem (sensor) can be arbitrarily reduced compared to its response when its dissipated heat becomes negative for a sufficiently large information flow to the other subsystem (demon). Such apparent violation of the second law, which is made possible by the nonreciprocal coupling between the two subsystems, is at the origin of enhanced nonequilibrium sensing. We illustrate this generic result with the example of two overdamped harmonic oscillators (Fig. 1).

## Harada-Sasa relation for subsystems

We begin by deriving a Harada-Sasa relation for coupled subsystems. We consider a composite system consisting of $d$ overdamped degrees of freedom $\mathbf{z}(t)$ in contact with a viscous equilibrium environment characterized by a temperature $T$ and a friction coefficient $\gamma$, whose dynamics obeys the Langevin equation (we set $k_B = 1$)[31]

$$\gamma \dot{\mathbf{z}}(t) = \mathbf{f}(\mathbf{z}(t)) + \sqrt{2\gamma T}\boldsymbol{\xi}(t), \qquad (1)$$

where $\mathbf{f}(\mathbf{z})$ are arbitrary forces acting on the system and $\boldsymbol{\xi}(t)$ is a vector of mutually independent Gaussian white noises. When the forces are nonconservative (for example, external driving forces or nonreciprocal interactions), the nonequilibrium steady state of the system is characterized by a positive rate of heat dissipation[16]

$$\dot{Q}_{\mathrm{diss}} = T\sigma = \langle \mathbf{f}^{\mathrm{T}} \circ \dot{\mathbf{z}} \rangle = \frac{1}{\gamma}\left\langle \| \mathbf{f} - T\boldsymbol{\nabla}_z \ln p_{\mathrm{st}} \|^2 \right\rangle_{\mathrm{st}} \geq 0, \qquad (2)$$

where $\circ$ is the Stratonovich product and $\langle ... \rangle_{\mathrm{st}}$ denotes the average with respect to the steady-state probability density $p_{\mathrm{st}}(\mathbf{x})$. The quantity $\sigma$ is the total entropy production rate that represents the increase in entropy of both the system and the environment due to the nonequilibrium nature of the dynamics[16]. We further divide the degrees of freedom into two subsets, $\mathbf{z} = (\mathbf{x}, \mathbf{y})$, and interpret $\mathbf{x}$ and $\mathbf{y}$ as the degrees of freedom of the subsystems $X$ and $Y$, respectively. Subsystem $X$ will be the sensor whereas subsystem $Y$ will act as the demon. Doing the same for the forces, $\mathbf{f} = (\mathbf{f}^X, \mathbf{f}^Y)$, we can split the total dissipation into local contributions from $X$ and $Y$,

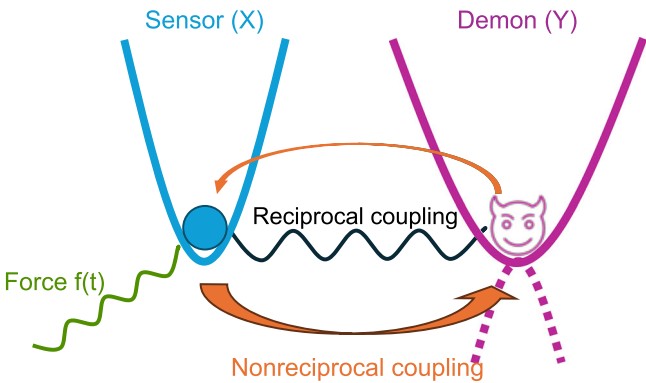

**Fig. 1 | Sensor-demon system.** The sensor consists of a Brownian particle ($X$) (blue) that is coupled via reciprocal (black spring) and nonreciprocal (orange arrows) interactions to another Brownian particle ($Y$) (purple) which acts as a demon. The demon increases the nonequilibrium signal-to-noise ratio by reducing the fluctuations of the sensor at the expense of its own. Optimal sensing of a force $f(t)$ (green) is achieved for an inverted harmonic potential for the demon (dotted).

$\dot{Q}_{\mathrm{diss}} = \langle \mathbf{f}^{X,\mathrm{T}} \circ \dot{\mathbf{x}} \rangle + \langle \mathbf{f}^{Y,\mathrm{T}} \circ \dot{\mathbf{y}} \rangle = \dot{Q}_{\mathrm{diss}}^X + \dot{Q}_{\mathrm{diss}}^Y$. To simplify the notation, we will focus on a two-dimensional space, $\mathbf{z} = (x, y)$, with single-variable subsystems.

By quantifying fluctuations of the velocity $\dot{x}(t)$ of subsystem $X$ (sensor) in a given frequency interval by the power spectral density $S_v^X(\omega)$ and its response to an external perturbation by the function $R_v^X(\omega)$[31], we obtain the local Harada-Sasa relation for subsystem $X$,

$$\frac{\gamma}{\pi}\int_0^\infty d\omega \left[ S_v^X(\omega) - 2TR_v^X(\omega) \right] = \langle f^X \circ \dot{X} \rangle = \dot{Q}_{\mathrm{diss}}^X, \qquad (3)$$

that connects the violation of the local fluctuation-dissipation theorem, $S_v^X(\omega) = 2TR_v^X(\omega)$[18,19], to the local heat dissipation rate $\dot{Q}_{\mathrm{diss}}^X$ (Methods).

## Improved nonequilibrium sensing

Equation (3) for the local subsystem has the same form as the global Harada-Sasa relation for the composite system[27–30]. However, the underlying physics is radically different. According to the global second law, $\dot{Q}_{\mathrm{diss}} \geq 0$, (2), the rate of heat dissipation is positive. The Harada-Sasa relation then implies that driving the system out of equilibrium always reduces the overall response compared to the fluctuations. This seems to suggest that better sensing, with a larger signal-to-noise ratio, is to be achieved near equilibrium. By contrast, the local second law for subsystem $X$ reads $T\sigma^X = \dot{Q}_{\mathrm{diss}}^X + Tl^X \geq 0$, where $\sigma^X$ is the local entropy production and $l^X = \left\langle \left( f^X - T\boldsymbol{\nabla}_x \ln p_{\mathrm{st}} \right)^{\mathrm{T}} \boldsymbol{\nabla}_x \ln p_{\mathrm{st}} \right\rangle_{\mathrm{st}}/\gamma$ is the so-called learning rate, which quantifies the information flow between the subsystems[21–25]. Through the action of the demon, the local heat dissipation rate $\dot{Q}_{\mathrm{diss}}^X$ of subsystem $X$ can become negative in the presence of a sufficiently large information flow $l^X$. This effect allows one to cool $X$ or to continuously extract work from it; it is the foundation for what has been termed nonreciprocal cooling[32–34]. A direct consequence of (3) is that the demon, with the help of the same effect, can also suppress the fluctuations of the subsystem compared to its response, and hence increase the signal-to-noise ratio.

The response function $R_v^X(\omega)$ in (3) is the real part of the complex response function, and therefore only accounts for the in-phase response of the velocity. In practice, we are often interested in the amplitude of the response, which is characterized by the absolute value of the complex response function[31],

$$\bar{R}_v^X(\omega) = \sqrt{R_v^X(\omega)^2 + \tilde{R}_v^X(\omega)^2}, \qquad (4)$$

where $\tilde{R}_v^X(\omega)$ is the imaginary part of the complex response function that measures the out-of-phase response of the velocity. Equation (4) can be used to define the dimensionless signal-to-noise ratio of the sensor

$$\mathrm{SNR}^X(\omega) = \frac{\bar{R}_x^X(\omega)f}{\sqrt{\mathrm{Var}_{\mathrm{st}}(x)}} = \frac{\bar{R}_v^X(\omega)f}{\omega\sqrt{\mathrm{Var}_{\mathrm{st}}(x)}}, \qquad (5)$$

where $f$ is the applied perturbation, $\mathrm{Var}_{\mathrm{st}}(x)$ is the variance of $x$ and $\bar{R}_x^X(\omega) = \bar{R}_v^X(\omega)/\omega$. The main result of this paper is that there is no fundamental upper limit on $\mathrm{SNR}^X$ away from equilibrium: in principle, we may design a system that has arbitrarily small fluctuations compared to the response, as we will now demonstrate in a concrete system. By contrast, the fundamental limit of the signal-to-noise ratio in an equilibrium system comes from the fact that noise cannot be eliminated at equilibrium: thermal fluctuations are indeed in general proportional to temperature[31], as can be seen from the expression of the Johnson-Nyquist noise in a resistor[35] and from the (generalized) equipartition theorem for nonharmonic confining potentials[35].

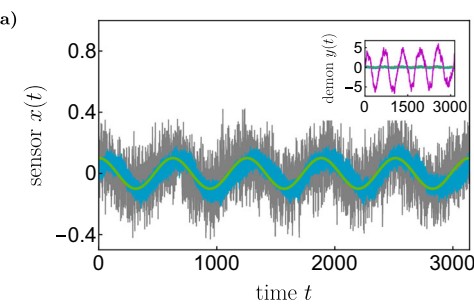

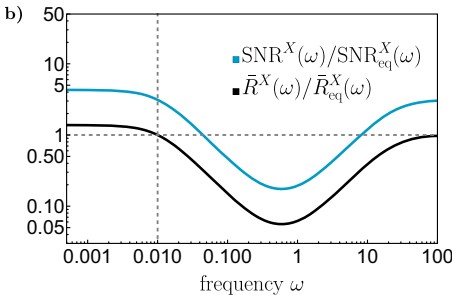

**Fig. 2 | Enhanced nonequilibrium sensing. a** The response of the sensor $x(t)$ to a periodic perturbation $\epsilon \cos(\omega_0 t)$ (green) exhibits less fluctuations in the presence of the demon (blue) than in equilibrium (gray). The inset shows the much larger fluctuations of the demon (purple). **b** Response function $\bar{R}_v^X(\omega)$, (4) (black), and signal-to-noise ratio, (5) (blue), normalized by their equilibrium values, which they both exceed below $\omega_0$ (vertical dotted line). Parameters are $\sigma = 10$, $T = 0.01$, $\omega_0 = 0.01$, $\epsilon = 0.1$ and $\gamma = 1$. Coupling parameters after minimization of the signal-to-noise ratio for a constant response are $k_x = 15.50$, $k_y = -7.919$, $\kappa = 8.269$, $\delta = -7.766$, corresponding to an eigenvalue $\lambda^- = 0.0105$ of the force matrix.

## Application to a linear system

Let us consider a two-dimensional system where subsystems $X$ and $Y$ can be locally approximated by linearly coupled harmonic oscillators (Fig. 1). The dynamics of the composite system follows the Langevin equation (1) with $\boldsymbol{f}(\boldsymbol{z}(t)) = -\boldsymbol{K}\boldsymbol{z}(t)$. We parameterize the force matrix $\boldsymbol{K}$ as

$$\boldsymbol{K} = \begin{pmatrix} k_x + \kappa & -\kappa - \delta \\ -\kappa + \delta & k_y + \kappa \end{pmatrix}. \quad (6)$$

This corresponds to two overdamped particles confined in parabolic traps with strengths $k_x$ and $k_y$. The particles interact via a spring with spring constant $\kappa$. In addition, the parameter $\delta$ describes a non-reciprocal coupling between the two particles. Similar nonreciprocal interactions have recently been realized experimentally in optically levitated particles[36]; they also naturally occur in active colloidal systems[37]. Since the dynamics is linear, we can analytically compute all the relevant quantities (Supplementary Information). The heat flow out of subsystem $X$ (sensor) is explicitly given by,

$$\dot{Q}_{\text{diss}}^X = \frac{2T\delta(\delta + \kappa)}{\gamma \mathcal{T}}, \quad (7)$$

whereas the corresponding response and variance read

$$\bar{R}_v^X(\omega)^2 = \frac{\omega^2 [\mathcal{Q}^2 + (\gamma\omega)^2]}{[(\lambda^+)^2 + (\gamma\omega)^2][(\lambda^-)^2 + (\gamma\omega)^2]},$$
$$\text{Var}_{\text{st}}(x) = \frac{T(\gamma\sigma + 2\mathcal{Q}) - \sqrt{\gamma\sigma \left[\gamma\sigma - 4\frac{(\mathcal{Q}-\lambda^+)(\mathcal{Q}-\lambda^-)}{\lambda^+ + \lambda^-}\right]}}{2\lambda^+ \lambda^-}, \quad (8)$$

where $\lambda^{\pm}$ are the eigenvalues of the force matrix $\boldsymbol{K}$. We further have the trace $\text{tr}(\boldsymbol{K}) = \mathcal{T}$, the determinant $\det(\boldsymbol{K}) = \mathcal{D}$, and $\mathcal{Q} = k_y + \kappa$. In order to warrant a stable steady state, we impose the condition $\mathcal{D} > 0$; from the inequality $\mathcal{T} \geq \sqrt{2\mathcal{D}}$, we then have $\mathcal{T} > 0$. The response function does not explicitly depend on the overall dissipation $\sigma = \dot{Q}_{\text{diss}}/T$, that is, on how far the overall system is driven from equilibrium, contrary to the variance. Therefore, for a given response function, the fluctuations can generally be reduced by driving the system out of equilibrium. We emphasize that both reciprocal ($\kappa \neq 0$) and nonreciprocal ($\delta \neq 0$) couplings are necessary to obtain a negative heat flow ($\dot{Q}_{\text{diss}}^X < 0$ for $-\kappa < \delta < 0$) and to achieve enhanced sensing with a reduced variance.

We next numerically illustrate the beneficial role of the demon on the sensor for a small periodic perturbation, $f(t) = \epsilon \cos(\omega_0 t)$, applied to the sensor. To that end, we set the amplitude of the nonequilibrium response at frequency $\omega_0$ to the corresponding equilibrium response,

$\bar{R}_v^X(\omega_0) = \bar{R}_{v,\text{eq}}^X(\omega_0)$, where $\bar{R}_{v,\text{eq}}^X(\omega) = \sqrt{(\gamma\omega)^2/k_x^2 + (\gamma\omega)^2}$ is the response spectrum of the sensor in the absence of the demon. We also fix the total rate of dissipation $\sigma$. Then, we numerically minimize the variance with respect to the eigenvalues $\lambda^+$ and $\lambda^-$, which gives us the least possible amount of fluctuations for a given response and dissipation.

Figure 2 a) displays the response of the sensor $x(t)$ in equilibrium, in the absence of the demon (gray), and with the demon (blue) to the small perturbation $f(t)$ (green) for the optimized parameters. A strong decrease in fluctuations is clearly visible; for the considered example, it amounts to a factor of 3.1 improvement in the signal-to-noise ratio. The behavior of the demon (purple) is shown in the inset for comparison; as discussed in more details below, it corresponds to an almost unstable mode that exhibits much larger fluctuations than the sensor. Figure 2b) moreover shows the response function $\bar{R}_v^X(\omega)$, (4) (black) and the signal-to-noise ratio $\text{SNR}^X(\omega)$, (5) (blue) relative to their respective equilibrium values, as a function of the frequency $\omega$. At frequencies lower than the reference frequency $\omega_0$, the response in the presence of the demon is enhanced, both in terms of its absolute value and its real part. At intermediate frequencies, the coupling to the demon reduces the response, while at high frequencies, where we essentially measure the viscosity of the environment, the response is unaffected.

## Fundamental sensing limit

To investigate the fundamental limit on the performance of the nonequilibrium sensor, we now consider the results of the above optimization of the sensor's parameters as a function of the reference frequency $\omega_0$, as shown in Fig. 3a). For frequencies above the characteristic relaxation rate $\omega_c = k_x/\gamma$ of the sensor, where response and fluctuations are governed by the properties of the environment rather than the system, no improvement is possible. By contrast, at low frequencies $\omega_0 \ll \omega_c$, the signal-to-noise ratio can be significantly enhanced above its equilibrium value. In particular, in the low frequency limit, where the equilibrium signal-to-noise saturates at a value of unity for the present parameters, the optimized nonequilibrium signal-to-noise ratio diverges as $\omega_0^{-1/4}$. This implies that, for sensing of low-frequency signals, in particular of constant forces, the amount of fluctuations can be decreased arbitrarily, while keeping the response and dissipation finite. We note that while Fig. 3a) shows the signa-to-noise ratio normalized by its equilibrium value, the latter approaches the constant $f/\sqrt{k_x T}$ in the low-frequency limit (Supplementary Information), so the divergence originates from an actual divergence of the nonequilibrium signal-to-noise ratio. Specifically, using the scaling of the parameters obtained from the numerical minimization, we find for given $\sigma$ and in the limit $\omega_0 \to 0$ (Supplementary

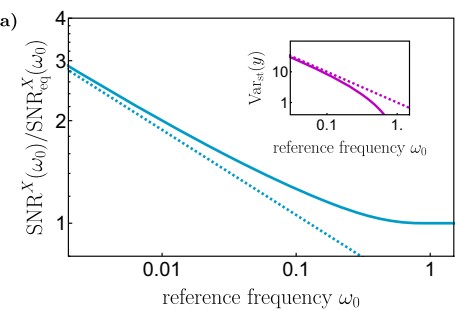

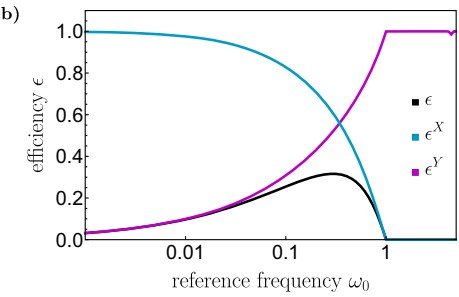

**Fig. 3 | Nonequilibrium sensing limit. a** The nonequilibrium signal-to-noise ratio, (5) (blue solid), exceeds its equilibrium value for low frequencies $\omega_0$, and diverges as $\omega_0^{-1/4}$ for $\omega_0 \to 0$, (9) (blue dashed), indicating the absence of a fundamental limit on out-of-equilibrium sensing. The result is obtained by fixing the response to the equilibrium response, $\bar{R}_v^X(\omega_0) = \bar{R}_{v,\,\mathrm{eq}}^X(\omega_0)$, corresponding to $k_x = 1$, with $\sigma = 1$, and then minimizing the variance with respect to the eigenvalues $\lambda^+$ and $\lambda^-$ of the force matrix. The divergence of the signal-to-noise ratio of the sensor is accompanied by diverging fluctuations of the demon (purple, inset). **b** The efficiency of the sensor (blue) approaches unity when the signal-to-noise ratio diverges for $\omega_0 \to 0$, indicating that the information acquired by the demon is perfectly converted into a negative heat flow from the sensor, However, the efficiency of the demon (purple) and the overall efficiency (black) vanish, indicating that the demon is a bad cooler in this limit. In general, the efficiencies of demon and sensor exhibit opposite behavior at low and high $\omega_0$.

Information),

$$\frac{\mathrm{Var}_{\mathrm{opt}}(x)}{\mathrm{Var}_{\mathrm{eq}}(x)} \simeq \sqrt{\frac{8\omega_0}{\sigma}}, \tag{9}$$

which agrees with the results obtained by explicit numerical optimization in the low-frequency regime.

To understand the origin of this dramatic improvement, it is useful to consider the optimal values of the eigenvalues $\lambda^{\pm}$. The increase in the signal-to-noise ratio is accompanied by a decrease of the smaller eigenvalue $\lambda^- \simeq \gamma \omega_0$. As a result, the stability of one of the eigenmodes of the system decreases. Since the response of the sensor is kept fixed, this eigenmode corresponds to the degree of freedom of the demon, which would be unstable, with negative spring constant, without the stabilizing coupling to the sensor. This instability allows the demon to absorb the fluctuations of the sensor, thus improving the corresponding signal-to-noise ratio at the expense of its own fluctuations, which grow as $\mathrm{Var}_{\mathrm{st}}(y) \simeq T/(\gamma \omega_0)$ in the low-frequency limit (Fig. 3a, inset).

### Information-thermodynamic efficiencies

Additional insight may be gained by examining the information-thermodynamic efficiencies of sensor and demon[21]

$$\epsilon^X = \frac{-\dot{Q}_{\mathrm{diss}}^X}{Tl^Y} \quad \text{and} \quad \epsilon^Y = \frac{Tl^Y}{\dot{Q}_{\mathrm{diss}}^Y}. \tag{10}$$

The parameter $\epsilon^Y$ quantifies how much information the demon acquires about the sensor relative to the amount of heat it dissipates, whereas $\epsilon^X$ is the efficiency of translating the acquired information into a negative heat flow that yields a reduction of the fluctuations of the sensor. The product $\epsilon = \epsilon^X \epsilon^Y = -\dot{Q}_{\mathrm{diss}}^X/\dot{Q}_{\mathrm{diss}}^Y$ is the overall thermodynamic efficiency of the combined system, that is, the relation between the heat removed from the sensor and the heat dissipated by the demon. All three quantities are displayed in Fig. 3b) as a function of the frequency $\omega_0$. We see that the dynamics of the sensor become approximately reversible ($\epsilon^X \to 1$ and $\sigma^X \to 0$) in the low-frequency limit in which the signal-to-noise ratio diverges. By contrast, the demon is not efficient in extracting information about the sensor ($\epsilon^Y \to 0$) in this limit, causing the overall thermodynamic efficiency $\epsilon$ to vanish. The constraint that the demon should reduce the fluctuations of the sensor while maintaining its response hence prevents it from acting as an efficient cooling device. This makes nonreciprocal sensing very different from nonreciprocal cooling[33,34].

## Discussion

We have investigated the physical limits on nonequilibrium sensing by analyzing a general sensor coupled to a demon. Our first key result is the identification of the central role of nonreciprocal sensor-demon interactions to enable the demon to significantly suppress fluctuations of the sensor, while keeping the response unaffected. As a consequence, the signal-to-noise ratio can be strongly enhanced compared to its equilibrium value. However, not just any non-reciprocal coupling will produce an improved sensor. Our second main result is to show that the parameters of sensor and demon must be properly optimized to achieve an arbitrarily large signal-to-noise ratio. Remarkably, it may even diverge at low frequencies in linear systems, revealing that there is no fundamental limit on none-quilibrium sensing. Our third nontrivial observation is that such divergent signal-to-noise ratio can be obtained with constant none-quilibrium entropy production, that is, with given energy dissipation. These findings should allow one to extend the applicability of enhanced nonequilibrium sensing to a wider context, beyond the limits of biology, where parameters are usually fixed, including physics and engineering. In particular, the possibility of enhanced sensing with constant energy dissipation/consumption appears to be an interesting property in the context of current research on energy-efficient wireless sensor networks where power minimization is critical[38–40]. Our predictions for coupled oscillators could furthermore be directly tested using optically levitated particles[36] or active colloidal systems[37]. We hasten to add that they also hold for systems described by discrete master equations for which there is no Harada-Sasa relation (Supplementary Information). Moreover, we stress that the fundamental requirement is nonreciprocal coupling between different degrees of freedom driving the overall system out of equilibrium; the separation into subsystems is convenient for intuition but not necessary. All in all, our work suggests that appropriately designed nonequilibrium systems might be generally used for highly accurate sensing, even in the presence of large environmental fluctuations.

## Methods

We here relate the local heat dissipation rates to the local fluctuations and responses of each subsystem. The fluctuations of the variable $z(t)$ can be quantified with the (positive definite) power spectral density matrix[31]

$$(S(\omega))_{kl} = \frac{1}{2} \int_{-\infty}^{\infty} dt \, e^{i\omega t} \langle \delta z_k(t) \delta z_l(0) \rangle + \langle \delta z_l(t) \delta z_k(0) \rangle, \tag{11}$$

with $\delta z(t) = z(t) - \langle z \rangle_{st}$. Its integral over all frequencies is equal to the steady-state fluctuations of $z(t)$, $\int_0^\infty d\omega \, (S(\omega))_{kl}/\pi = \langle \delta z_k \delta z_l \rangle_{st}$. That is, $S_z(\omega)d\omega$ measures the amount of fluctuations of $z(t)$ in the frequency interval $[\omega, \omega + d\omega]$. A closely related quantity is the velocity power spectral density matrix, $S_v(\omega) = \omega^2 S(\omega)$, which likewise measures the fluctuations of $\dot{z}(t)$ in a given frequency interval[31]. On the other hand, the response of $z(t)$ to a perturbation force $\eta\phi(t)\hat{e}_l$ applied in direction $l$ can, to linear order in the magnitude $\eta$ of the perturbation, be expressed as[31]

$$\langle z_k(t) \rangle^\eta - \langle z_k \rangle_{st} \simeq \eta \int_0^t dt' \int_0^{t'} dt'' \, \mathcal{R}_{v,kl}(t' - t'')\phi(t''), \quad (12)$$

where $\langle \ldots \rangle^\eta$ denotes the average evaluated in the perturbed system and the matrix $\mathcal{R}_v(t' - t'')$ is the velocity-response matrix, whose components measure how much the velocity in direction $k$ at time $t'$ changes in response to an applied force in direction $l$ at time $t''$. Note that, due to causality, $\mathcal{R}_v(t' - t'')$ is only defined for $t' \geq t''$. Real and imaginary parts of the frequency-response matrix are given by $R_v(\omega) = \int_0^\infty dt \cos \omega t \, \mathcal{R}_v(t)$ and $\tilde{R}_v(\omega) = \int_0^\infty dt \sin \omega t \, \mathcal{R}_v(t)$.

To simplify the notation, we proceed by focusing on a two-dimensional space, $z = (x, y)$, with single-variable subsystems. Then, we can write the two matrices

$$S_v(\omega) = \begin{pmatrix} S_v^X & S_v^{XY} \\ S_v^{XY} & S_v^Y \end{pmatrix} \text{ and } R_v(\omega) = \begin{pmatrix} R_v^X & R_v^{XY} \\ R_v^{YX} & R_v^Y \end{pmatrix}, \quad (13)$$

where $S_v^X(\omega)$ and $S_v^Y(\omega)$ are the respective velocity power spectral densities of $X$ and $Y$, and $S_v^{XY}(\omega) = S_v^{YX}(\omega)$ quantifies the correlations between the two subsystems. Similarly, $R_v^X(\omega)$ measures the response of system $X$ to perturbations applied to itself, while $R_v^{XY}(\omega)$ measures the response of system $X$ to perturbations applied to $Y$. Out of equilibrium, the response is generally not reciprocal, $R_v^{XY}(\omega) \neq R_v^{YX}(\omega)$.

Using the explicit expressions for $S_v$ and $R_v$, we obtain the local Harada-Sasa relation for subsystem $X$ (sensor),

$$\frac{\gamma}{\pi} \int_0^\infty d\omega \left[ S_v^X(\omega) - 2TR_v^X(\omega) \right] = \left\langle f^X \circ \dot{X} \right\rangle = \dot{Q}_{\text{diss}}^X, \quad (14)$$

that connects the violation of the local fluctuation-dissipation theorem, $S_v^X(\omega) = 2TR_v^X(\omega)$[18,19], to the local heat dissipation rate $\dot{Q}_{\text{diss}}^X$ (Supplementary Information). A similar relation holds for subsystem $Y$ (demon).

## Data availability

No datasets were generated or analysed during the current study.

## Code availability

The `Mathematica` code used for the optimization of the SNR and plotting the figures will be provided upon request.

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

## Acknowledgements

A.D. is supported by JSPS KAKENHI (Grant No. 22K13974, 24H00833 and 25K00926) and JST ERATO Grant Number JPMJER2302. E.L. would like to thank the Physics Department at the University of Kyoto, where part of this work was carried out, for their hospitality. He further acknowledges support from the DFG (Grant FOR 2724).

## Author contributions

A.D. and E.L. conceived and designed this study. A.D. calculated and optimized the SNR. A.D. and E.L. wrote the manuscript.

## Funding

## Competing interests

The authors declare no competing interests.
