## [Transparent Peer Review file · Nature Communications]

Fundamental limits on nonequilibrium sensing

Corresponding Author: Dr Andreas Dechant

Version 0:

Reviewer comments:

Reviewer #2

(Remarks to the Author)

I find the reply of the authors to my previous comments to be satisfactory. Regarding their reply to the comments of the first referee, I agree with the authors in that the Harada-Sasa relationship is only useful insofar as it provides an intuitive understanding of the observed effect, but that is not really necessary for the derivations of the results, since all the quantities involved can be evaluated independently. This can be done for both continuous and discrete systems, as the authors show in the example added to the supplementary material. The authors argue that the limited state-space of discrete systems prevents the unbounded increase of the signal-to-noise ratio, which might deserve a deeper analysis but is out of the scope of the present manuscript.

Overall, I think the manuscript reports an interesting observation that might lead to new developments in stochastic thermodynamics. I am inclined to recommend its acceptance in Nature Communications.

Reviewer #3

(Remarks to the Author)

The results presented in the manuscript are interesting but I think the answer to my main question remain unclear.

1) What is the fundamental limit on the SNR in an equilibrium system? What are the universal quantities that would show up on the right hand side of such fundamental limit?

2) What is the same limit on a nonequilibrium system (full system not a subsystem in the presence of another subsystem like a Maxwell demon)?

3) Same question for an equilibrium subsystem?

4) Is it correct that the authors have shown that there is no fundamental limit on the increase of the SNR, as compared to the SNR of an equilibrium system, in a nonequilibrium subsystem in the presence of a Maxwell demon for small frequencies ω_0 ? In other words it is not the SNR that has a fundamental limit that is violated but rather the ratio $\text{SNR}/\text{SNR}_{\text{eq}}$ that can be arbitrarily high for low enough frequencies?

5) I do not think the limitation their Harada-Sasa to continuous systems is really a limitation of their paper. Their result would survive even if they never mentioned the relation: they did show the "lack of a limit" in an example. They do not really need their Harada-Sasa relation for this Main Result.

6) Concerning the Harada-Sasa relation and the "necessity" of negative dissipation for larger response. This is true for the particular situation they are looking at but it is not true in general since the relation is not valid for discrete systems.

7) My view is that the authors have found a situation where the SNR can be arbitrarily increased, in relation to its equilibrium value, in the regime of w_0 going to zero. There might other non-equilibrium setups when this happen, including the case of a full system without the need of the presence of a Maxwell demon. It is not a violation of limit in the SNR rather the quantity here is the ration SNR/SNR_{eq} . I do not think this situation is clearly explained in the manuscript. However, I do find their result potentially interesting.

Version 1:

Reviewer comments:

Reviewer #3

(Remarks to the Author)

I am satisfied with the answers to my questions. The modifications in the manuscript are also appropriate. I recommend publication.

Third reply to Reviewer #3:

0) *The results presented in the manuscript are interesting but I think the answer to my main question remain unclear.*

We are grateful to the Referee for the repeated review and the helpful remarks.

1) *What is the fundamental limit on the SNR in an equilibrium system? What are the universal quantities that would show up on the right hand side of such fundamental limit?*

The fundamental limit of the SNR in an equilibrium system comes from the fact noise cannot be eliminated at equilibrium: thermal fluctuations are indeed in general proportional to temperature. For a resistor R , it is for example given by the Johnson-Nyquist expression: $\langle V^2 \rangle = 4k_B RT \Delta f$ where Δf is the frequency bandwidth. On the other hand, for a nonharmonic potential of the form x^s , it is given by the (generalized) equipartition theorem: $\langle H_{pot} \rangle = k_B T/s$. Both cases are, for instance, discussed in the recent book by Merhav ‘*Statistical Physics for Electrical Engineering*’ (Springer, 2018), Ref. [32].

For the specific situation considered in the manuscript (two overdamped degrees of freedom X and Y), the best possible SNR in equilibrium is realized when X is not coupled to Y . In other words, in equilibrium, coupling the sensor to an auxiliary system cannot improve its SNR. Concretely, the equilibrium SNR of X is bounded by [see Eq. (S72) in the Supplementary Information]:

$$SNR_{eq}^X(\omega) \leq \sqrt{\frac{k_x}{k_x^2 + (\gamma\omega)^2}} \frac{f}{\sqrt{T}}$$

where the right-hand side is the SNR in the absence of coupling between X and Y . Here, f is the magnitude of the input signal, k_x is the trapping strength of X , γ is the friction coefficient and T the temperature.

We now indicate after Eq. (5) that: “By contrast, the fundamental limit of the signal-to-noise ratio in an equilibrium system comes from the fact noise cannot be eliminated at equilibrium: thermal fluctuations are indeed in general proportional to temperature [31], as can be seen from the expression of the Johnson-Nyquist noise in a resistor [32] and from the (generalized) equipartition theorem for nonharmonic confining potentials [32].”

2) *What is the same limit on a nonequilibrium system (full system not a subsystem in the presence of another subsystem like a Maxwell demon)?*

The division into subsystems is only for convenience of interpretation. We could equivalently well view the system and the demon together as a single nonequilibrium

system, in which case, the results of the manuscript apply to the overall nonequilibrium system. We have added the following remark to the discussion to clarify this point:

“Moreover, we stress that the fundamental requirement is nonreciprocal coupling between different degrees of freedom driving the overall system out of equilibrium; the separation into subsystems is convenient for intuition but not necessary.”

3) Same question for an equilibrium subsystem?

For both equilibrium and nonequilibrium systems, the division into subsystems is entirely arbitrary and does not change the underlying physical properties. So, the limit of the SNR for an equilibrium subsystem does not differ from the limit of a general equilibrium system.

4) Is it correct that the authors have shown that there is no fundamental limit on the increase of the SNR, as compared to the SNR of an equilibrium system, in a nonequilibrium subsystem in the presence of a Maxwell demon for small frequencies ω_0 ? In other words it is not the SNR that has a fundamental limit that is violated but rather the ratio SNR/SNR_{eq} that can be arbitrarily high for low enough frequencies?

In Fig. 3, the ratio between the nonequilibrium and equilibrium SNR is shown for simplicity to obtain a dimensionless quantity. However, since the SNR for the equilibrium system approaches a constant in the limit $\omega_0 \rightarrow 0$ (see the expression in the answer to question 1), it is the actual magnitude, rather than just the ratio, of the nonequilibrium SNR that diverges. We have added the following remark before Eq. (9) to clarify this point:

“We note that while Fig. 3a) shows the signal-to-noise ratio normalized by its equilibrium value, the latter approaches the constant $\frac{f}{\sqrt{k_x T}}$ in the low-frequency limit (Supplementary Information), so the divergence originates from an actual divergence of the nonequilibrium signal-to-noise ratio.”

5) I do not think the limitation their Harada-Sasa to continuous systems is really a limitation of their paper. Their result would survive even if they never mentioned the relation: they did show the "lack of a limit" in an example. They do not really need their Harada-Sasa relation for this Main Result.

We thank the Referee for their remark. Indeed, as we now argue in the manuscript and explicitly demonstrate in the Supplementary Information, the same approach also applies to discrete systems where there is no Harada-Sasa relation.

6) Concerning the Harada-Sasa relation and the "necessity" of negative dissipation for larger response. This is true for the particular situation they are looking at but it is not true in general since the relation is not valid for discrete systems.

As remarked above, we agree with the Referee that the Harada-Sasa relation is not necessarily related to larger response. However, at least for the system investigated in the manuscript, indeed the regime of enhanced response corresponds to an apparently negative dissipation for the system, as is stated in the paragraph following Eq. (8). This is reasonable, since we optimize the SNR by reducing the fluctuations of the sensor at fixed response. Lower fluctuations correspond to a reduced effective temperature of the system, causing it to absorb heat from the environment.

7) My view is that the authors have found a situation where the SNR can be arbitrarily increased, in relation to its equilibrium value, in the regime of w_0 going to zero. There might other non-equilibrium setups when this happens, including the case of a full system without the need of the presence of a Maxwell demon. It is not a violation of limit in the SNR rather the quantity here is the ration SNR/SNR_{eq} . I do not think this situation is clearly explained in the manuscript. However, I do find their result potentially interesting.

As we remarked in the reply to question 4, it is not merely the ratio between the non-equilibrium SNR and the equilibrium SNR that diverges, but rather the actual SNR of the nonequilibrium system. As correctly pointed out by the Referee, it is the non-equilibrium nature of the system that enables this rather than the (arbitrary but useful for intuition) subdivision into system and demon.